# How Beyond-5G and 6G Makes IIoT and the Smart Grid Green—A Survey

**DOI:** 10.3390/s25134222

**Published:** 2025-07-06

**Authors:** Pal Varga, Áron István Jászberényi, Dániel Pásztor, Balazs Nagy, Muhammad Nasar, David Raisz

**Affiliations:** 1Department of Telecommunications and Artificial Intelligence, Faculty of Electrical Engineering and Informatics, Budapest University of Technology and Economics, H-1111 Budapest, Hungary; 2Department of Automation and Applied Informatics, Faculty of Electrical Engineering and Informatics, Budapest University of Technology and Economics, H-1111 Budapest, Hungary; 3Department of Electric Power Engineering, Faculty of Electrical Engineering and Informatics, Budapest University of Technology and Economics, H-1111 Budapest, Hungary

**Keywords:** green networking, 5G, 6G, smart grid, energy efficiency, edge computing, industrial IoT, sustainability

## Abstract

The convergence of next-generation wireless communication technologies and modern energy infrastructure presents a promising path toward sustainable and intelligent systems. This survey explores how beyond-5G and 6G communication technologies can support the greening of Industrial Internet of Things (IIoT) systems and smart grids. It highlights the critical challenges in achieving energy efficiency, interoperability, and real-time responsiveness across different domains. The paper reviews key enablers such as LPWAN, wake-up radios, mobile edge computing, and energy harvesting techniques for green IoT, as well as optimization strategies for 5G/6G networks and data center operations. Furthermore, it examines the role of 5G in enabling reliable, ultra-low-latency data communication for advanced smart grid applications, such as distributed generation, precise load control, and intelligent feeder automation. Through a structured analysis of recent advances and open research problems, the paper aims to identify essential directions for future research and development in building energy-efficient, resilient, and scalable smart infrastructures powered by intelligent wireless networks.

## 1. Introduction

The ongoing global push toward sustainability and energy efficiency is reshaping the design and operation of modern communication systems and power infrastructures. With climate change mitigation as a critical global priority, both the telecommunications and energy sectors are under pressure to reduce their environmental footprint while continuing to meet rapidly growing demand. As a result, the convergence of next-generation wireless technologies with smart grid architectures is emerging as a key enabler of greener, more adaptive, and more intelligent infrastructures.

The Industrial Internet of Things (IIoT), driven by massive-scale sensor deployments, plays an important role in transforming traditional energy grids into intelligent and responsive smart grids. However, the exponential growth in connected devices and data traffic has also raised concerns about power consumption, scalability, and network sustainability. Beyond-5G and 6G technologies offer a unique opportunity to address these challenges [1] by providing ultra-reliable low-latency communication, high device density support, and integrated intelligence to optimize energy usage dynamically [2].

This survey is motivated by the lack of consolidated knowledge at the intersection of energy-conscious communication technologies and intelligent energy infrastructures. By mapping the current state of the art and identifying critical challenges and gaps, this paper aims to provide a foundation for researchers and practitioners aiming to create future-proof, energy-efficient systems that align with the goals of Industry 5.0 and global sustainability objectives.

Recent advancements in beyond-5G and 6G technologies promise substantial enhancements in the energy efficiency and adaptive intelligence of smart grid systems. Despite this potential, several unresolved challenges hinder the full integration and optimal use of these technologies in IIoT and smart grid contexts. This paper aims to address these challenges and highlight critical research gaps that may currently limit the advancement of green communication systems and smart energy infrastructures.

One of the key issues is the fragmented approach to energy efficiency optimization across network layers. While considerable efforts have focused on physical-layer improvements, such as low-power wide-area networks (LPWANs) and energy harvesting techniques [3,4], fewer studies have thoroughly investigated cross-layer solutions. Integrated frameworks that encompass network management, edge computing, and artificial intelligence-driven optimization remain underexplored [5,6].

Another gap exists in creating unified architectures that seamlessly combine IIoT, smart grid operations, and next-generation wireless networks. The existing literature typically treats these components separately, neglecting the critical interdependencies and specific real-time requirements inherent in energy applications [7,8]. Consequently, comprehensive assessments of end-to-end system sustainability and standardized benchmarks for deployments are clearly lacking.

Furthermore, the balance between energy efficiency and security remains inadequately addressed. Current low-energy communication protocols frequently compromise robustness and data integrity to minimize power consumption. This reveals a clear gap in designing secure, reliable, and energy-efficient communication systems suitable for critical infrastructure, particularly in the highly dense and heterogeneous device environments anticipated in beyond-5G and 6G networks [9,10].

Standardization and interoperability also present ongoing challenges. The absence of unified protocols and standards complicates IIoT device deployment and operation across diverse networks and energy grid architectures, causing inefficiencies and excessive energy usage [3,8]. This underscores the need for cross-domain standards and protocols to simplify implementation and maximize efficiency.

Finally, although AI-driven solutions for energy management have gained significant attention, most studies remain theoretical or rely on oversimplified simulations. There is an acute shortage of empirical studies and testbed validations demonstrating these solutions’ effectiveness under practical conditions and dynamic operational scenarios of smart grids [6,11,12].

Through a structured examination of the current literature and critical analysis of these issues, this survey aims to identify key interdisciplinary gaps. It aims to find routes toward holistic, efficient, and robust integration of beyond-5G and 6G communication technologies into the smart grid, setting clear directions for future research.

Our key contributions through this survey are as follows:We provide a comprehensive review of the role of beyond-5G (B5G) and 6G wireless communication technologies in supporting sustainable Industrial Internet of Things (IIoT) and smart grid infrastructures.We identify and classify the main challenges in achieving energy efficiency, real-time responsiveness, and interoperability in smart energy systems powered by next-generation mobile networks.We examine major technological enablers for green IoT, including low-power wide-area networks (LPWAN), wake-up radios, energy harvesting methods, and edge computing architectures.We analyze energy-aware optimization strategies in 5G/6G networks and data centers, with a focus on AI-driven orchestration and resource management.We map critical smart grid applications to their communication requirements and discuss how emerging wireless capabilities—such as ultra-reliable low-latency communication and massive device connectivity—enable these use cases.We highlight existing research gaps, particularly in the areas of cross-layer energy optimization, empirical validation, and the integration of secure communication with energy efficiency.We propose future research directions to address the lack of unified standards and the need for interoperable, scalable, and testbed-validated communication platforms for sustainable cyber-physical infrastructures.

There are two recent surveys partially covering the area that we are investigating. To better understand the distinguishing areas and approach of our paper, the similarities and differences are clearly shown by Table 1. The uniqueness is already shown by the scope.

The next section describes various green IoT approaches in the context of 5G and 6G networks. Section 3 summarizes the green core and telco cloud strategies for reducing energy consumption. Section 4 elaborates on the integration and application of green communication technologies within smart grid systems, highlighting their practical implications and use-cases. Finally, Section 5 discusses conclusions and outlines future research directions.

## 2. Green IoT in 5G and 6G Networks

While IoT (Internet of Things) covers a wide range of devices, they are generally understood to be an electronic device capable of interacting with the physical environment and transmitting digital data over the Internet, providing a connection between the physical and digital world. There are numerous application areas for IoT devices, such as food supply chains, smart homes and cities, and transporting industries [16]. While the numbers vary based on the surveys, there are currently around 15.4 billion IoT devices, and predictions point to around 30 billion devices by 2030 [17].

When discussing IIoT (Industrial IoT), people mostly refer to manufacturing and production, but in wider terms, it also covers the IoT solutions for the whole supply chain—including transportation and vehicular technologies, the energy sector, and sustainable approaches such as circular economy [18]. Opening up the picture, this does not only mean field devices in manufacturing plants but also their data sharing, processing, and feedback control to MEC (Manufacturing Execution Control) and ERP (Enterprise Resource Planning) for the whole value chain. Industry 4.0 is based on this IIoT idea [18], extending information exchange to MEC, ERP, and even external service consumers. Industry 5.0 is the latest round of innovations with native use of AI solutions, sustainability, and cooperative human–machine solutions [19]. These systems utilize 5G and beyond capabilities in various ways [20,21], so their diverse set of requirements (such as high endpoint density, or ultra-reliable low-latency communication (URLLC) are met).

On the energy utilization side of the picture, research into the current climate change has resulted in a gradual shift towards renewable energy sources. Due to the current and projected number of IoT devices, energy usage has become the focus of green IoT and IIoT, which aims to reduce the carbon footprint of such devices by increasing energy efficiency and renewable energy usage [22].

There are important key sustainability metrics and KPIs used to evaluate green IoT systems. These include energy efficiency (bits/Joule), carbon footprint (CO_2_ equivalent emissions per device or network segment), energy consumption per device or per application, energy harvesting efficiency, and embodied energy (total energy consumed during manufacturing and deployment). Further KPIs used in different domains are sleep mode efficiency, resource utilization rate, and green cost per bit, which balances energy and economic efficiency [23]. Together, these metrics enable the quantitative assessment of environmental impact, guide energy-aware system design, and support regulatory compliance in sustainable IoT and IIoT deployments.

By 2029, 5G is expected to be available to over 85% of the global population [24]. While previous generations of mobile communication networks focused mainly on increasing bandwidth by improving spectral efficiency (abbreviated as SE), energy efficiency (EE) has also become a focal point to lower energy usage in the future for 5G deployment.

It is very important that 5G has special requirements for machine-to-machine communication, describing slices for URLLC, massive IoT, and enhanced mobile broadband (eMBB). The first two of these have a large number of industrial IoT use cases, and hence, the development and deployment of 5G are inseparable from the success of IIoT and Industry 4.0 [25].

This section will examine different techniques to reduce energy consumption within 5G cellular networks.

### 2.1. LPWAN: Enabler for Ultra-Low-Energy 5G

Among the IoT infrastructures, there are networks where one node generates small information pieces and sends them periodically, and these nodes are confined to a smaller area, such as warehouse inventory management or environment monitoring devices (usually called wireless sensor networks). In these cases, integrating a 5G (or any other generation) transmitter into every node is infeasible due to the size and cost constraints. By having only one central node in this network equipped with a 5G transmitter and larger energy sources, we can enable low-cost devices to transmit data over 5G through this gatewaynode while also providing similar latency as 5G.

A LPWAN(low-power wide-area network) is a wireless communication network focusing on low energy usage, long communication distances, and low costs. There are several competing standards under this name, the most notable being Sigfox, LoRa, and NB-IoT. The bandwidth varies between 0.3 kbit/s and 1Mbit/s, while the communication range can reach up to 40 km, depending on the environment and technology used [3].

In [8], three of the most popular LPWAN implementations are compared for several different properties (SigFox, LoRa, and NB-IoT). While SigFox and LoRa use unlicensed frequency bands, NB-IoT operates within licensed frequency bands, making it one of the more expensive options while providing significantly better performance, latency, and scalability. For these reasons, NB-IoT is mainly aimed at industrial use.

LoRa and SigFox have similar properties. LoRaWAN provides three different classes based on the latency requirements, which generally outperforms SigFox. In turn, SigFox provides a larger coverage area (around a 40 km radius compared to LoRaWAN’s 20 km).

One major difference is the deployment of networks. SigFox uses proprietary base stations, and only partners licensed with the company (called SigFox Operators) are allowed to deploy these stations. Consumers must pay a subscription fee to their local operator to use their network. For LoRa, Semtech owns the IP (intellectual property) for the radio modules. While Semtech manufactures its own modules, it also licenses out to other manufacturers, such as ST Microelectronics and Microchip. With LoRa, one can deploy their own private networks or subscribe to a network provider. Table 2 compares these networks by other properties.

Due to its openness, LoRaWAN has received more interest and backing from industry partners. It was examined for the green IoT environment in [26]. They evaluated and found that slotted Aloha scheduling for delay-tolerant applications outperforms pure Aloha in terms of collisions and throughput, endorsing it for green IoT. If the application is delay-sensitive, pure Aloha scheduling is recommended.

In [27], the authors compared LoRa and SigFox for energy consumption under different firmware update scenarios. They found that while both are competent solutions for IoT communication, LoRa used less energy with the trade-off of having a smaller coverage area.

### 2.2. Wake-Up Radio: Rethinking Sleep Cycles

In most battery-powered IoT devices, sleep cycles (also known as duty-cycled medium access control) are used to switch the integrated circuits into sleep mode, drastically reducing their power consumption. For example, a micro-controller from Microchip Technology (ATmega328P, widely used in IoT boards such as Ardunio) [28], consumes around 6 mW power at idle (when the supply voltage is 3 V and the clock frequency is 4 MHz). In contrast, when in power-saving mode (PSM), the chip only consumes 2.4 μW power.

Figure 1 shows the current draw of an NB-IoT module during different phases. It can be seen that in contrast to the IDLE phase, there is no communication during the PSM phase, so the current draw can be low for the module.

The microcontroller typically works in cycles to save power. While the chip is usually in sleep mode, an RTC (Real-time Clock) provides a periodic interrupt to wake up the microcontroller.

These cycles inherently come with a trade-off: Longer cycles lower the energy usage and increase the latency of the system. The node also wakes up periodically even when no processing is needed, further increasing energy usage.

To combat these disadvantages, research has focused on wake-up radios [30]. These are low-power (typically in the range of sleep-state power draw) radio modules capable of receiving wireless messages and waking up the processing unit. The use of such a radio eliminates unnecessary wake-ups while also decreasing the latency of communication. This mode is advantageous when a central station queries the nodes without any periodicity.

A wake-up radio has been implemented on a testbed with 5G in mind [6]. The wake-up receivers are made from some basic electronic components (resistors, diodes, capacitors, coils, and one nanopower comparator). Meanwhile, the transceiver is based on an nRF52 development kit, with its purpose being a gateway between the mobile application (communicating through Bluetooth) and the wake-up-enabled devices. Through measurements, they have found that the receiver module draws around 1.2 μW power, comparable to the sleep state of the microcontroller. They have also found that wake-up-enabled IoT devices have a higher lifetime and lower communication latency.

Recently, 3GPP incorporated WUS (Wake-Up Signal) in Release 16 [31], which allows the device to know whether there are pending transactions left, waking up the processor unit only when needed. Given the time frame, no measurements conducted around 5G compatible wake-up modules, but it is expected to provide the same power-saving benefits as general wake-up radios.

In 6G networks, wake-up radios will play an even bigger role due to the large number of base stations in ultra-dense networking. In [32], the authors propose a fuzzy logic-based wake-up strategy to prevent base stations from constantly switching between different energy states. This logic is further aided by the traffic volume of the network as well as the solar energy generated near the modules.

### 2.3. Energy Harvesting

IoT devices can reduce energy consumption by implementing a sleep mode or turning off the radio, but this is not enough for a long or potentially indefinite operating cycle. By harnessing energy from the surrounding environment, such as solar, thermal, or kinetic energy, IoT devices can become self-sufficient and reduce the reliance on traditional electrical power sources. This not only improves their efficiency but also enables deployment in remote or hard-to-reach areas where access to power may be limited.

There are many energy-harvesting solutions that can be used for IoT. One of the most popular energy harvesting solutions is photovoltaic panels, which convert sunlight into electricity. There are other feasible options, such as thermal energy harvesters, which use temperature differences to generate electric power, and electromagnetic and kinetic energy harvesters, which generate energy from motion or vibration.

The feasible harvesting methods and their advantages and disadvantages are investigated in in [4,33]. None of the examined solutions are capable of providing constant and reliable power or high power density. In particular, solar energy is highly dependent on environmental factors, making it impossible to use in places where solar coverage is low or where dust can easily settle on still surfaces. Kinetic energy harvesters are prone to failures due to their repetitive motion, while thermal-based solutions typically provide low efficiency. Some solutions have been proposed to mitigate these effects [34].

Given all this, it is advised to use an onboard battery supply to store the generated energy. In some cases, when high reliability is not mandatory, the carbon footprint can be further reduced by removing the battery, optionally replacing it with a simpler capacitor. In this case, the system can have frequent and unpredictable power outages, and there is no fully reliable intermittent computing solution for IoT systems [35].

### 2.4. Device-to-Device Communication

Device-to-device (D2D) communication occurs in a network when two devices communicate with each other without the help of a base station. Among many its advantages, D2D communication can increase data throughput and provide communication even when mobile communication networks are down (useful in emergencies), significantly reducing communication time and allowing a more reliable connection (depending on the environment). The power draw is also lower due to direct communication, making it optimal for green communication networks.

D2D communication has been heavily researched. In particular, Zhang et al. [36] proposed several power allocation schemes to maximize the throughput of a communication network constrained by energy use, making it ideal for green IoT communication networks.

While in previous generations of mobile cellular networks, D2D communication has not been properly used, it is predicted to play a significant role in 5G. In [37], many different implementations and prototypes of D2D have been examined. The results show numerous benefits of using D2D in 5G cellular networks while also outlining some challenges, such as interference and resource allocation.

When multiple devices connect to each other through D2D, a network can be established between them. These types of networks are called *ad hoc networks*, and they have been researched extensively [38]. In [39], ad hoc networks were investigated to reduce energy usage in 5G. The paper focuses on green data centers providing data services for mobile devices. The authors found that their proposed methods can significantly improve energy efficiency.

D2D communication will also play a major role in 6G networks. In super-dense, high-throughput networks, shorter multi-hop communication is favored due to the higher reliability of the transmission. The authors in [40] focus on efficient communication in 6G networks with D2D.

Another major challenge with device-to-device communication is security. Both devices have to identify the other party and use proper encryption to avoid man-in-the-middle attacks and eavesdropping. A survey has been conducted around the types of attacks faced by 5G D2D communications [41], and it showed defensive strategies to combat these attacks. In green communication networks, great care must be taken to find the right balance between security and power consumption.

### 2.5. MU-MIMO: Massive Antenna Arrays

Cellular networks often use *Multi-User Multiple-Input and Multiple-Output* (MU-MIMO) techniques to increase data throughput, channel robustness, and signal-to-noise ratio. This is achieved by having multiple antennas on the transmitting and receiving side and using a data signal processor on the receiver sides to reconstruct the original signals. While the previous cellular generation (LTE) supported up to eight transceiver antennae in a base station, 5G will extend this above 100 individual antennas in a base station, arranged in an array grid. Due to the number of antennas, this is often called *Massive MIMO*.

Massive MIMO systems have been regarded as a key enabling technology for future networks [42]. With massive MIMO, the array grid is cheaper due to the amplifiers’ low cost compared to the traditional ultra-linear, high-power amplifiers. It has also been shown that massive MIMO drastically increases both SE and EE in multiple scenarios [43]. There are further emerging technologies for 6G, including eXtremely Large-scale MIMO (XL-MIMO) [44], Intelligent Reflecting Surfaces (IRS), and Cell-Free mMIMO (CF-mMIMO) [45] that allow further energy optimization.

In [46], a more detailed energy model of 5G MIMO communication is given, and different simulations are run. With this model, they have found an optimal number of antennas to maximize the system’s energy efficiency with regard to the number of users. The authors also investigated the energy efficiency of a hybrid massive MIMO system, where other 5G technologies are also present (such as mmWave and HetNets), and found numerous benefits, such as easier beamforming, which can further increase energy efficiency and reliability, as can be seen in Figure 2.

### 2.6. Mobile Edge Computing

As information technology has evolved over the last few decades, increasing computational power and energy efficiency, so has the complexity of applications running on mobile devices. For example, a modern mobile phone typically comes with some form of face recognition software installed, which requires heavy image processing. While the devices currently available are perfectly capable of performing these tasks, running them depletes the battery faster. This increases the number of recharges (which also produces energy loss) and the degradation of the battery.

Mobile Edge Computing (MEC) has been proposed to offload computationally intensive tasks. MEC provides a computing environment near the base station to allow local task execution, decreasing latency, network congestion, and energy usage. The specification also allows authorized third parties to run their application on this environment, further extending the number of use cases, such as DNS and content caching.

In [47], the authors presented a detailed overview of MEC, focusing mainly on communication. They found that while MEC provides numerous benefits, it also brings some new challenges that need to be further researched, such as network security and deployment issues. MEC is put in the perspective on IIoT and Industry 5.0 in [48], where the authors argue that telco-grade solutions (such as telco-grade MEC) make service guarantees more achievable than earlier architectures. The authors in [49] also look at edge computing in the context of beyond-5G and 6G networks. Regarding 6G, the authors of [50] propose an architecture angled toward green MEC services to be achieved by programmable data planes.

When using MEC, the energy requirement of the network must also be investigated. This has been accomplished in [51], where a general power model was established for mobile communication systems.

To improve the energy efficiency of mobile devices, a task offloading scheme was proposed in [52]. The authors introduced a system model on which a problem is formulated and then proposed an Energy-Efficient Computational Offload (EECO) scheme, which minimized energy usage while fulfilling pre-defined delay constraints. To achieve this, every mobile device was first classified based on its tasks, and a priority was assigned to each. With the priorities known, radio resources were allocated to the devices to offload their tasks. Compared to the scheme without any offloading, EECO achieved, on average, 18% lower energy usage.

### 2.7. Further Support for Green IoT in 6G

Reconfigurable Intelligent Surfaces (RIS) and Non-Orthogonal Multiple Access (NOMA) are two transformative technologies in 6G that significantly enhance the energy efficiency and scalability of green IoT systems [53].

RIS refers to programmable metasurfaces [54,55] that can manipulate electromagnetic waves to improve signal propagation. There are passive and active RIS approaches [56]. Passive RIS reflects incident signals without additional power consumption, using programmable meta-surfaces to control phase shifts for enhanced signal propagation. Active RIS, in contrast, incorporates amplification or signal processing components to boost or regenerate signals, enabling improved coverage and performance at the cost of higher energy usage. In IoT environments, where devices are often energy-constrained and deployed in challenging locations, RIS can passively reflect and steer signals without consuming much power. This reduces the required transmission power from IoT devices and base stations, leading to extended device lifetimes and lower overall network energy consumption [57].

NOMA is a multiple access technique proposed for B5G and 6G mobile networks [58] that enables multiple users to simultaneously share the same time–frequency resources by leveraging power-domain multiplexing and successive interference cancellation. Unlike orthogonal schemes that allocate distinct resources to each user, NOMA superimposes signals with different power levels at the transmitter, allowing users with better channel conditions to decode and subtract weaker users’ signals before decoding their own [59]. This approach significantly enhances spectral efficiency, supports massive connectivity, and reduces latency, making it particularly advantageous for IoT applications where heterogeneous devices with varying power capabilities coexist.

In short, from an IoT perspective, NOMA enables multiple IoT devices to share the same frequency–time resources by superimposing signals at different power levels. By doing so, it improves spectral efficiency and allows massive numbers of IoT devices to be served simultaneously. From a green IoT perspective, NOMA reduces idle time and control overhead, which in turn minimizes unnecessary power usage. Moreover, it allows low-power IoT devices to operate with reduced energy by leveraging stronger channel conditions of nearby devices through cooperative communication, aligning with the energy and sustainability goals of green 6G networks.

When RIS and NOMA are combined, their synergy can be particularly powerful for green IoT networks. RIS can be used to enhance the channel conditions of weak IoT links, making NOMA decoding more reliable and less energy-intensive. This not only improves connectivity for devices in obstructed environments (e.g., factories, underground sensors) but also reduces the power demands on each device. As a result, the joint use of RIS and NOMA can support massive IoT deployments with ultra-low energy budgets, which is essential for sustainable smart cities, precision agriculture, and remote environmental monitoring [60].

Furthermore, the integration of RIS and NOMA in 6G aligns with edge intelligence trends by enabling context-aware energy management. For example, AI-based controllers at the network edge can dynamically reconfigure RIS and NOMA power levels based on real-time IoT traffic, device battery status, or environmental factors. This leads to self-optimizing, energy-aware IoT systems with minimal human intervention. Altogether, RIS and NOMA are not just physical-layer enhancements but key enablers of scalable and sustainable green IoT ecosystems in the new 6G era.

### 2.8. Lessons Learned

In this section, numerous techniques were presented to improve the energy efficiency of 5G networks. Reducing energy consumption has been one of the main driving forces of the last decade, ushering in new classifications, such as green IoT and green 5G. Five key areas have been identified in this section, which will most likely have an important part in future communication networks: low-power wide-area networks, wake-up radio, device-to-device communication, MU-MIMO, and Mobile Edge Computing. I would also like to present some important papers summarizing the future of green 5G networks.

The authors in [61] have researched five interconnected areas with regard to green 5G. They analyzed the differences between classical Shannon theory and systems in real environments by optimizing spectral and energy efficiency for both and compared the SE and EE ratio between different communication networks. They have also investigated different properties of the network architecture, such as rethinking area divisions by cells, investigating signaling and control signals, and examining the advantages of irregular antenna arrays and full duplex radios.

A more complete survey can be found in [62]. They investigated a plethora of different subjects. Starting with the comparison of previous cellular network generations, they introduce different techniques for energy allocation. Different scenarios are investigated for energy efficiency, such as D2D communication and ultra-dense networks. The survey also shows different security and privacy issues related to power optimization.

## 3. Green Core and Telco Cloud

It is important to strive to use green energy sources and reduce consumption by cutting energy waste. Although the use of renewable energy sources is gaining ground, it cannot replace the use of organic energy sources due to growing consumption needs. Worldwide consumption from green energy sources, as shown in Figure 3, represents less than 15% of the energy produced [63]. Supplying energy from renewable energy sources is not always feasible, as electricity generation is not a building block of the network infrastructure, but energy reduction should be.

The core network of the fifth-generation cellular networks is very different from previous generations: 5G core networks are cloud-native by design, so 5G runs on regular data center hardware. The age of vendor-specific hardware is mostly over. The power consumption of Data Centers (DC) is huge. In a Data Communication Network (DCN) environment, not just the servers but also the network equipment consumes a high amount of power. The largest energy consumer is the cooling of the aforementioned equipment. Figure 4 shows the typical consumption of Data Centers. Estimated global DC electricity consumption in 2022 was 240–340 TWh, which is 1–1.5% of global electricity use [65]. The cooling of the servers can reach similar amounts of energy as the operation of the servers, which is why many operators are building their new DCNs close to the Arctic [66]. There is a very high financial incentive to reduce the power consumption of Data Center networks; thus, researching this subject is very lucrative for researchers, and research material is plentiful. There are several ways to make the Cloud greener, as described below.

### 3.1. Making the Server’s Hardware Components Use Less Electricity

This is a very straight-forward way to make the Cloud greener, but the server hardware industry is only researched by a very few manufacturers who can produce DC hardware. For example, there are two leading companies manufacturing DC-grade processors—AMD and Intel [68]—and there are two companies manufacturing DC-grade HDDs—Western Digital and Seagate [69]. There has been an immense amount of progress in this field, but independent research materials are mostly lacking. A few papers are available discussing the efficiency of AMD Zen CPUs, like [70], and Intel Xeon CPUs, like [71]. Traditionally, x86-based platforms dominate, but concerns over their power draw have opened the door to alternative architectures and new processor designs. For instance, a trial by NTT Docomo and NEC demonstrated that Arm-based Graviton2 server processors consumed 72% less energy than equivalent x86 chips for a 5G core workload [72]. Reducing processor consumption is an important research area, but only a few sets of published results are available. In [73], the authors discuss an efficient deep idle core power-state architecture called AgileWatts (AW) that improves the power efficiency of CPUs used in DCs with low performance degradation. In [74], the authors compare the HDD and SSD RAID solutions’ impact on server consumption. The study shows that because of the different characteristics, different storage solutions are suitable for different scenarios to achieve energy efficiency. Beyond CPUs, offloading specialized tasks from general-purpose cores to dedicated accelerators is gaining traction. Data processing units (DPUs) and SmartNICs can handle packet processing or encryption more efficiently than CPUs, reducing overall server power consumption [75].

Most research and improvement in this field focuses on making DCNs greener. Given that the maximal consumption of a single component is power-constrained due to cooling, e.g., a single socket in a 2U server has not been allowed to consume more than 225 W for a decade now. Thus, manufacturers have to increase processing and storage capacity without increasing power consumption. AMD-TSMC produced the most innovation in this field in the past few years; they developed an affordable, highly energy-efficient server processor family, the AMD Epyc [70,76,77]. Intel is also working on lowering consumption. The Intel 4th Gen Xeon processors provide an energy-efficient, high-performance solution specifically for 5G User Plane Function (UPF) [78].

### 3.2. Making Data-Center Cooling More Power Efficient

The cooling subsystem is the single most power-consuming subsystem of most Data Centers. Cooling consumes between 30 and 50% of the total power consumption of the DCs [79]. To make cooling more power efficient, modern designs are adopting both advanced hardware and intelligent control strategies [80]. Many cooling solutions are available for DCs, with different energy savings rates and promising efficiency [81].

There are three main approaches for cooling optimization in modern data centers. The first tends to maximize the efficiency of cooling equipment through technological optimization, like [82]. The aim of the second approach is to optimize the control policy of active cooling systems [83]. The third is the High-Temperature Data Center concept, which tests the highest optimal ambient temperature at which devices can still operate reliably [84]. The latter two solutions can also be used to optimize already operational data centers.

The energy used to cool data centers should be optimized, but it does not always have a negative impact on the environment, as it is possible to avoid wasting heat by using the produced thermal energy. The heat generated can be used to heat surrounding buildings, reducing heating demand and costs [85].

Paper [86] highlights how telecom companies are adopting green data centers to reduce energy consumption and environmental impact while maintaining operational efficiency. Key innovations include integrating renewable energy sources, advanced cooling technologies like liquid immersion and free-air cooling, and AI-driven energy management systems to optimize performance and sustainability. These strategies, combined with modular designs and edge computing, position the telecom industry to meet growing digital demands while significantly reducing its carbon footprint.

### 3.3. Integrating Green Energy Sources into the Power Network of the Data Center

Only a small portion of the globally produced and used energy is from renewable energy sources (Figure 3). Data Centers are not responsible for energy production, but since they use a large amount of it, they can be energy self-sufficient. The integration of renewable energy sources into existing DCNs does reduce the carbon footprint. This is also a highly researched subject because green energy production is highly erratic, while the power consumption of Data Centers is mostly deterministic. The research on this subject is much more colorful than the hardware research. A wide variety of approaches have been considered, including the following:Adding non-electric cooling [87]Pre-cooling [88]Usage of high-capacity batteries [89]Thermal storage to increase the capability to consume energy from renewable sources [87]Making better and fairer bargaining algorithms to further increase renewable sources [90]Making use of geo-distributed Data Centers to load-balance to those DCNs that have renewable energy at their disposal at a given moment [91]Usage of solar-power plant architecture that works well with DCNs [92]Off-site green power through Power Purchase Agreements (PPAs) or utility partnerships [93,94]

As a result, the industry’s share of energy from renewables has been rising, nearly 20% of telco energy consumption in 2023 was supplied by green sources, roughly double the share in 2019 [95].

### 3.4. Optimizing the Virtual-Hardware Resource Allocation

Nowadays, most virtual machine instances are completely separate from the actual hardware that runs the virtual instance. Direct virtual-hardware resource allocation (CPU pinning) is very rarely used; most of the time, virtual machines are load-balanced on many CPUs on the fly. Thus, we have the opportunity to load-balance instances to minimize power consumption [96]. The power consumption of a single instance can vary immensely, with factors including the following: type of processor, use frequency, load of the rest of the processor, temperature of the processor, and time required to complete the calculation [97]. Load-balancing does not require hardware development or DCN infrastructure to develop and validate scheduling and load-balancing algorithms because there are very well-developed tools to simulate the power usage of Clouds [98,99]. On the other hand, this is a very well-researched subject.

There are research studies that propose decreasing overall consumption by increasing the amount of overall resources [98]. Resource allocation can be based on prediction as well [100]. Another solution is utilizing real-world energy prices and green energy availability to use DCs that have cheaper renewable energy at their disposal [101,102]. This is a very good approach to decreasing the carbon footprint of data centers because very little hardware investment is required, and it can be used with existing infrastructure. It also makes operators highly interested in using green energy resources to reduce their total electricity costs.

A key strategy is to maximize resource utilization on active servers and turn off or idle the rest. To address this, researchers are developing energy-aware orchestration frameworks that treat energy as a first-class optimization metric alongside throughput and latency. One example is Ericsson’s cloud-native core solution that adds an “application-aware” shim layer to Kubernetes, allowing the platform to apply aggressive power management tailored to each network function’s needs [103]. Another vein of research focuses on algorithmic optimization of VNF placement and scheduling. For instance, a deep reinforcement learning framework (GreenNFV) was proposed to continuously tune resource shares (CPU cores, frequency, cache allocation, etc.) for running network functions [104].

### 3.5. Optimization Beyond the Core Network

On the network element level, minimizing power consumption can also be achieved by (self-)monitoring resource needs and applying predictive algorithms, as mentioned above. The actions taken based on these are either decreasing the usage of power-intensive tasks or completely going to sleep until a given trigger arrives (including a timeout). Energy Management and Monitoring Applications (EMMAs) can help mobile networks estimate fronthaul and backhaul energy consumption and trigger reactions [105].

On the service level, power optimization has an effect on end-to-end operation, and the respective actions are based on service usage analysis. The first types of decisions to use fewer (or, if needed fewer) resources are seasonal. Examples include increased eMBB coverage during match hours inside and around stadiums, during business hours around office estates, or during the tourist seasons around resorts. The resource coverage of these areas should be decreased outside the busy time periods, which leads to decreased power consumption if done right (i.e., turn off the radio, suspend machines in the Cloud, or use them for other services). Public network usage data is limited; therefore, the number of studies on this topic is very limited.

While improving the core network and data center efficiency is vital, a holistic approach to green networking extends beyond the core to other domains of the telecom network, notably the Radio Access Network (RAN) and transport network. In mobile networks, the RAN is the largest energy consumer, typically accounting for about 70–80% of total network energy usage [106]. Therefore, any comprehensive energy optimization must include aggressive strategies in the RAN [107].

On the application level, by writing code that requires less computational power, the power consumption of the application can be drastically reduced. This is a hard subject because there is no clear best software, but there are studies that propose techniques to improve energy efficiency. Green and sustainable software thinking can be integrated into agile scrum development [108]. On the topic of the state of the research into green and sustainable software [109], there are more than 500 publications.

### 3.6. A Summary on Minimizing Power Consumption in 5G

Minimizing power consumption at all levels and segments of the 5G infrastructure is an important goal of telecommunication system vendors and operators alike. This issue must be addressed both on the network element level and the service level.

On the network element level, minimizing power consumption can be reached through the (self-)monitoring of resource needs and even by applying predictive algorithms. The actions taken based on these are either decreasing the usage of power-intensive tasks or completely going to sleep until a given trigger arrives (including a timeout).

On the service level, power optimization has an effect on end-to-end operation, and the respective actions are based on service usage analysis. The first types of decisions to use fewer (or whether it is necessary to use fewer) resources are seasonal. Examples include increased eMBB coverage during match hours inside and around stadiums, during business hours around office estates, or during the tourist seasons around resorts. The resource coverage of these areas should be decreased outside busy time periods, which leads to decreased power consumption if done right (i.e., radio OFF and machines in the Cloud OFF or used for other services).

Table 3 conclusively summarizes the various techniques used to reduce energy consumption in 5G networks.

### 3.7. Towards 6G

The 6th generation mobile network is not available yet, but there are already many studies about it. Increasing energy efficiency—partially leading to sustainability—is also a major focus of the standardization process [110,111].

6G distinguishes itself from previous generations by embedding energy efficiency as a foundational design principle rather than a post-hoc optimization. Unlike 5G, which primarily emphasized data rate and latency improvements, 6G integrates energy-aware protocols, AI-native control mechanisms, and hardware-level innovations such as energy-efficient antennas and low-power integrated circuits. Moreover, 6G architectures are expected to support dynamic power management through intelligent orchestration of network elements, edge computing, and the use of RIS to reduce energy consumption in signal propagation.

To enhance energy efficiency in these high-complexity systems, the introduction of Artificial Intelligence (AI) is almost inevitable [112]. Reference [113] presents a comprehensive survey of AI techniques in resource management to enhance the performance and sustainability of 6G networks. AI algorithms will allow for real-time monitoring of patterns of utilization and expected demands on network resources in order to achieve energy-efficient resource management. This will enable dynamic resource allocation, reducing power consumption without sacrificing performance. Paper [114] presents a green 6G network architecture and protocol stack, integrating AI technology to optimize wireless network performance and resource allocation.

In-network computing is a new energy-efficient computing technology introduced to change the traditional CPU-centric computing model, which can be used in 6G networks [115]. The previous mobile network generations were designed under the assumption that the wireless environment could not be controlled or modified, but 6G makes a new concept possible. With the introduction of non-orthogonal multiple access (NOMA) beamforming and intelligent reflective surfaces (IRS), greater energy efficiency can be gained [116].

Sustainability gets special attention in 6G. It is not only about energy efficiency, but the whole lifecycle associated with the technology transformation is affected, equipment production to disposal, focusing on issues surrounding electronic waste, energy consumption, and environmental impact [117,118]. To achieve sustainability goals in 6G, it is essential to adopt circular economy strategies that promote recycling, reuse, and resource efficiency across the entire lifecycle of network infrastructure. Additionally, integrating extended producer responsibility and leveraging renewable energy and advanced recycling technologies can significantly reduce the environmental footprint of 6G deployments [119].

The deployment timeline for 6G is projected to span from early testbed experimentation in the mid-2020s to commercial rollout beginning around 2030. However, widespread adoption will be influenced by regulatory and standardization challenges, including spectrum allocation in sub-THz and THz bands, electromagnetic exposure limits, and cross-domain coordination between energy and telecommunications sectors. Regulatory bodies such as the ITU and regional authorities (e.g., ETSI, FCC) must develop harmonized frameworks that ensure interoperability, data privacy, and cybersecurity within smart grid-telecom integrations.

### 3.8. Challenges and Limitations of Green Technologies in 5G and Beyond Networks

One of the primary concerns in adopting energy-efficient communication protocols is the potential compromise in network security and data integrity. Many green techniques can seriously weaken the system’s ability to detect intrusions or ensure secure data transmission. As an example, lightweight protocols may lack robust encryption or authentication mechanisms, making them more vulnerable to attacks. Another example is aggressive sleep scheduling or duty cycling, which might delay or miss the detection of anomalies [120]. This could be particularly problematic in critical applications like smart grids or industrial automation where reliability and trust are key issues. This means we need balanced solutions that integrate adaptive security measures without disproportionately increasing energy costs.

While MEC and AI-driven orchestration are powerful enablers of energy savings as discussed, they introduce notable computational and control overhead. MEC nodes, often deployed at cell edges or near users, must continuously process and analyze large volumes of context data to make energy-aware decisions [121]. This local computation, while reducing core network load, can itself be energy intensive and may negate some of the intended savings. Moreover, training and running AI models at the edge poses additional challenges in terms of model accuracy, inference delay, and the sustainability of AI computational resources, especially when scaled across dense deployments [14].

Furthermore, the wide deployment of green infrastructure elements, such as energy-efficient base stations, edge servers, or energy harvesting modules, involves significant upfront investment. Although these technologies may reduce operational costs in the long run, the capital expenditure (CAPEX) required for initial rollout and integration—especially in underserved or rural areas—can be prohibitive. Additionally, the heterogeneous nature of edge environments makes standardized deployment difficult, often requiring custom integration or localized adaptations that further raise costs. These economic barriers may slow the adoption of green 5G infrastructure and widen the digital divide if not addressed through strategic incentives or public–private collaboration.

Finally, integrating multiple green technologies across layers can increase the overall system complexity. This complexity makes interoperability across vendors and legacy systems more challenging and may introduce management overhead or configuration errors that offset energy gains. As many green strategies require tight synchronization and real-time adaptation, which are difficult to maintain across distributed and heterogeneous devices. The lack of common standards or benchmarking tools for evaluating energy efficiency further complicates comparative assessments and widespread adoption.

Industrial IoT environments face similar but uniquely demanding challenges. In more and more cases, factory floors, robotic systems, and process automation pipelines rely on 5G and beyond, especially under time-sensitive networking (TSN) constraints. In such settings, the application of energy-efficient communication strategies must not compromise latency guarantees, reliability, or safety. Furthermore, IIoT deployments frequently involve heterogeneous devices with varying levels of processing capability and power budgets, complicating the uniform application of green techniques [25]. Energy harvesting or edge intelligence deployment may not be viable for legacy PLCs or deeply embedded devices. In IIoT as well, the increased control overhead of energy-optimized orchestration can burden constrained networks and introduce additional points of failure. This again highlights the need for industrial domain-specific green design principles and practices.

### 3.9. Lessons Learned

In this section, the energy efficiency of the 5G core network has been investigated to understand the energy requirements of the different elements of a telecom core network, which are based on a data center infrastructure. The aim of the study was to identify areas where energy consumption can be optimized without compromising network performance. The results showed that computing and cooling systems account for a significant share of the total energy consumption, highlighting the potential for implementing energy-efficient solutions in these areas. The study also explores the possibility of implementing energy-harvesting technologies. Table 4 summarizes the key concepts to make the core network more energy efficient.

As can be seen with the development of mobile networks, energy reduction is playing an increasing role in communication systems. While in the past the aim was to maximize performance, there is now a growing need to consider efficiency and hence energy savings when developing devices. We have now reached the point where device development can achieve a lower level of efficiency, and the focus has turned to optimization. Optimization at the software level remains an important aspect of development, and research into this will continue to be a major focus. The biggest step forward is in the use of the environmental factors. Energy is produced from renewable energy sources in the ambient environment, and efforts are also being made to recycle the energy residues that are generated, such as the heat produced.

Besides, there are some trade-offs, such as security vs. enery efficiency, or handling the overhead of MEC and AI at the edge—which are upcoming engineering challenges to tackle and optimize.

## 4. Green Data Collection for the Smart Grid

Current 5G communication and smart electricity grids are two different domains that can mutually interact and play an important role in reducing pollution by increasing the share of renewable generation in the electricity supply. This chapter presents slices of smart grid and 5G technology and how they can boost green power.

### 4.1. The Concept of the Smart Grid

The Smart Grid is a modern power system concept that integrates energy distribution control with information sharing, making electricity distribution more efficient than ever before [122]. In a layered approach, it follows the usual IoT architecture, consisting of perception, networking, and application layers [7], with special attention placed on security [123].

The EU Technology Platform has defined the Smart Grid as an intelligent electricity network able to integrate the actions of all users connected to it (generators, consumers, or both) in order to efficiently deliver sustainable, economical, and secure electricity supply. Similarly, the US Department of Energy describes the Smart Grid as an energy system that incorporates digital technology to improve the reliability, security, and efficiency of the electric system through information exchange, distributed generation, and storage resources for a fully automated power delivery network [124].

Recent research into the Smart Grid also includes the development of the Internet of Energy (IoE) either in exploring the convergence of the energy and information domains [125] or, more fundamentally, the transferring of electricity in multiple frequencies like internet data packets [126].

Other common defining characteristics include

integration of renewable energy sources, resulting in bidirectional energy flows;a high level of observability, enabled by applying large numbers of sensors and processing large volumes of data with high data rates;a high level of controllability by deploying actuators and power flow control devices;a shift from centralized to decentralized and distributed intelligence in distribution system automation;the increased engagement of customers to interact in the production–consumption balancing process, which includes demand-side management programs, direct load/generation/storage control by system operators, or price signal-driven manual or automated actions.

These (mostly challenge-driven but also technology-pushed) innovations are transforming the electricity system at all levels and require new operation concepts (e.g., new self-healing protection concepts, islanding, and microgrids) as well as the adoption of market design (including appropriately defined actors, market products, and regulations).

As a consequence, these all require robust telecommunication infrastructure for addressing reliability, capacity, and security issues that may potentially be settled by 5G technology [124,127].

To cope with these challenges, and while 6G technology (i.e., 5G and beyond) is still in the early stages of research and development, ITU-R has developed three schemes of 5G use cases covering Enhanced Mobile Broadband (eMBB), Ultra-Reliable and Low-Latency Communications (URLLC), and Massive Machine-Type Communications (mMTC) [128], that have been discussed in the previous section of this paper.

Since 2018, telecommunication companies have started implementing 5G technology in responding to the increasing needs of industrial and consumer customers for data interconnection [129]. This section investigates telecommunication network requirements, especially in terms of any possible roles of 5G and IIoT infrastructure in the smart grid domain.

The above sophisticated features present consequences to adopting advanced ICT infrastructure, especially in terms of latency, connection density, throughput, and security. Low-latency and ultra-reliable communication are two fundamental requirements of most smart grid operations, as the stability of their operation relies on digitally continuous coordination [130,131,132]. Technically, the smart grid allows two-way communication between central controllers and local actuators within the network. Through this, the smart grid should also enable supply units to quickly respond to urgent situations, such as the sudden increase of electric demand in certain areas, so that the power network remains stable [133]. Figure 5 illustrates the general architecture of the smart grid.

### 4.2. Smart Grid Data Flows

To understand the role of 5G in the smart grid domain, we introduce typical use cases of data and communication characteristics on the smart grid. As a smart system, it manages broad sources of data or signals from power generation, transmission, distribution, and consumption in high resolution using sensor networks [133]. The data is the processed for various adjustments, synchronizations, compensations, warnings, and analytics [134]. There are several use cases of smart grid applications that require wireless communications and are likely to be enabled by the 5G network. These include low-voltage distribution system data acquisition, intelligent distributed feeder automation, millisecond-level precise load control, and distributed power supply [128,135]. The following discussion is based on these use cases.

#### 4.2.1. Low-Voltage Distribution System Data Acquisition

Advanced Metering Infrastructure (AMI) is considered the most important in low-voltage distribution data acquirement. Data transmitted by smart meters provides energy-related information to distribution network operators, traders, and customers. The information on electricity consumption as well as voltage, current, and frequency is required to understand power quality as well as consumer behavior (including peak time and demand patterns) [136,137].

Furthermore, customers might want to form a local energy hub. A house connected to the smart grid may have some solar panels to generate electricity as sunlight can be caught everywhere, not to mention wind and other green resources. This house may sell the energy when the supply for itself is surplus, such as when the owners are working outside during the day.

As consumers may want to manage energy based on the best price status and decide whether to consume or to sell electricity, smart meters, together with other smart grid sensors discussed next, should be seen as big data sources from which the power companies can extract information on the patterns of demand, consumers’ behavior, effects of dynamic pricing, faults, etc. This approach is called big data analytics [138]. Information and decisions may be inaccurate if the network connection is unreliable, and a lack of the mentioned features may lead customers to lose their interest. Although requirements of low latency and high bandwidth may be tolerable, key features of successful communication in this category include high reliability as well as high concurrency of massive access [127,139,140].

#### 4.2.2. Intelligent Distributed Feeder Automation

The societal costs of electricity outages are comparable to those of severe natural disasters and run into billions of USD annually worldwide. Most of the outage frequency and duration figures stem from the distribution networks [141]. Therefore, sophisticated protection schemes based on significantly more sensors and connectivity at this level are key to a secure and affordable economy.

Intelligent Distribution Automation (IDA) employs automatic switching devices to improve the reliability and quality of power supply, providing high-quality services to users at low operational costs. This set of functionalities (FLISR: Fault Localization, Isolation/self-healing and Service Restoration) is considered most critical in distribution networks as it requires real-time communication between sensing and actuating. These functionalities are integrated with the SCADA (Supervisory Control And Data Acquisition) that comprises the basic information management system to ensure visibility and command transmission.

In addition, Unmanned Aerial Vehicles (UAV) have recently been employed for power grid inspection purposes. This also requires reliable and low-latency communication, as it inspects almost humanly unreachable systems, such as high-voltage transmission lines, and verifies them very carefully. Although this system utilizes small bandwidth and little device concurrency, it is characterized by ultra-low latency and high reliability of communication [142,143].

#### 4.2.3. Millisecond-Level Precise Load Control

The next form of robust wireless communication is found in millisecond-level precise load control (MLC). As introduced in [144,145], MLC not only observes power grid faults in real-time, but it also incorporates various power stations, energy storage stations, and interruptible loads in its control. As a typical example, when a fault occurs in high-voltage transmission, MLC will quickly disconnect less-important loads such as vehicle charging stations or other non-continuous production power supplies in factories [130,146]. Another use case for MLC is energy management in a microgrid or local energy community, either for the purpose of balancing and scheduled operations or in order to provide frequency control reserves for higher voltage level networks. Among the MLC devices, Phasor Measurement Units (PMU) are considered essential in Smart Grid operations [147,148,149].

A PMU is a device that detects the magnitude and phase angle of an electrical phasor quantity in the electricity grid using a centralized time source provided by a global positioning system (GPS) where the IEEE1588 Precision Time Protocol standard is used. The range of applications of PMUs is still the subject of intensive research, spanning areas from event analysis, tuning and validation of dynamic system models, monitoring of various oscillation modes, as well as stability margin monitoring or state estimation, targeting areas like supporting synchronization or service restoration, automated closed-loop protection systems (WAMPAC—Wide-Area Measurement, Protection, Automation, and Control).

PMUs are expected to be able to sample time-stamped phasors to be transmitted to a local or remote receiver at rates of up to 120 samples per second. In traditional grids, PMUs are generally installed across the transmission lines and used for synchronizing power sources from different generators. Failing to synchronize generators or larger areas, including several power sources, leads to instabilities and power outages. Recently, micro-PMUs (uPMUs), which are smaller versions of PMUs, have increasingly been applied at distribution systems, especially when operating bidirectional power flows like in smart grids. The uPMU is capable of storing and analyzing data locally or communicating it in real-time to grid operations so that it allows faster and precise time-stamping measurement supported by GPS [150]. As a consequence, ultra-low-latency and ultra-reliable communications are a necessity.

#### 4.2.4. Distributed Generation

Another typical case of wireless communication in smart grids is related to distributed power generation (DG). Since a smart grid allows its users to share their own electricity, energy flowing in the distribution lines is bidirectional and changes dynamically in real time. This means that connecting those distributed power supplies to power distribution networks also leads to new technical challenges in the secure and stable operation of the power distribution networks. One of these challenges is the synchronization of microgrids after a disconnection (the reason for which could be a short circuit that led to tripping).

Microgrids could span medium-sized geographical areas, ranging from several households supplied by one LV line to several settlements in a region. Such systems are likely to include several DG units as well as storage elements. The (re)synchronization process of such a microgrid to the public utility grid requires the adjustment of the phase angles of all DG units in the microgrid, which in turn requires low-latency and medium-to-high bandwidth for real-time operation.

A distributed power monitoring system that consists of master stations, sub-stations, and distributed terminals can be used to monitor and control the operation of distributed power generators automatically.

However, it requires widespread connection of devices and high reliability with medium-latency communications [128]. Table 5 distinguishes the four use cases’ communication requirements as described by various survey- and overview-type sources. It also summarizes how 5G meets current IIoT needs in four smart grid use cases, while 6G introduces transformative capabilities like AI-native networking and sub-millisecond responsiveness, enabling next-generation autonomous energy systems [144,151,152,153,154,155,156,157].

### 4.3. Data Communication Infrastructure in Smart Grid

Like network segmentation of the internet topology, the communication network in a smart grid is mainly divided into three main areas: Home Area Network (HAN), Neighborhood Area Network (NAN), and Wide Area Network (WAN) that will be described below [143,157,158].

HANHAN facilitates two-way communications to provide demand response in the smart grid. It transmits energy data from home appliances such as charging stations, light dimmers, and water heater control to a smart meter as a gateway to the next network. This network is commonly deployed within a house or an office and utilizes a relatively lower data rate (100 kbps) than the other two, with up to 15 s latency [159,160]. Broadband internet connection is commonly used while its bandwidth resource is shared with the house occupants through a wired or wireless modem. Either way, it can also use power line communication (PLC), ZigBee, Bluetooth, or other narrowband technologies. Another importance of HAN is that it can also be used for home automation, such as heat scheduling and smart switching of either lighting, garage door, etc., so that it leads to efficient home energy management.NANNAN, which is located between HANs and WAN, is the main bridge between smart meter infrastructure devices and utility operators. It transfers data bi-directionally either from HAN to data concentrators in WAN, or vise-versa. A NAN is operated within dense residential areas spanning several kilometers, and accumulatively requires a data rate transmission up to 1 Gbps, as it ultimately carries thousands of HANs’ data traffic. Multiple technologies can be used to construct this network.Broadband PLC is commonly used, as it offers reasonably cheap installation. However, due to flexibility and other limitations mentioned previously, it is quite often extended or replaced by wireless technologies. Various wireless-based technologies operating on a license-free spectrum like LoRa are used in some cases. However, in most other cases, cellular networks such as a Universal Mobile Telecommunication System (UMTS) or Long-Term Evolution (LTE) are used due to the reliability and higher data rate requirement, although 802.15.4-based wireless technology such as ZigBee is sometimes used as well [161,162].WANWhile HANs and NAN cover smaller areas, WAN covers the whole interconnection among main power generations, power transmissions and distributions, operators and controls, and local data concentrators. It delivers real-time data for M2M control and measurements and integrates several NANs. Moreover, the communication of all smart grid components, including the operator control center, main and renewable energy generation, transmission, and distribution, are finally interconnected based on WAN. Therefore, WAN has a very high transmission data rate of up to a few Gbps [143,163].

Various types of data link layers have been used to support those area networks. Power Line Communication (PLC) seems to be the closest one to this domain as it uses the power wire itself for its physical link, and therefore, it offers low cost and simple installation in a limited existing power line area. However, channel distortion and complex routing issues make it non-applicable to covering other parts of smart grid communications. Fiber optic cables are seen as the most promising medium for Smart Grid communications due to their unique advantages, such as high data rate, low latency and jitter, and the longest distance capability among wired communications. The only disadvantage is that they are very costly for massive connections, although fiber optic power cables are gradually becoming popular for long-distance point-to-point communications, especially at high transmission levels [122,163].

Wireless communications, on the other hand, offer significant benefits such as low-cost installations, rapid deployment, easy user access, and mobility. Recent studies in [152,164] indicate that there is a growing need for wireless data exceeding wired data technologies due to the transformation of the traditional grid to a Smart Grid. It is mentioned that there are many existing wireless network technologies that can be used to enable the Smart Grid, such as ZigBee, LoRa, Sigfox, Bluetooth, WiFi, terrestrial microwave, cellular, WiMax, and satellite communications. Each of these technologies has its own applications, advantages, and disadvantages.

Nevertheless, cellular communication technology is considered the best option due to its cost-effective infrastructure sharing, network coverage of large geographical areas, high data speed, sufficient security features, and reliability, as it uses a licensed band. Moreover, this technology has also been widely adopted in most countries and, therefore, is regarded as the key facilitator to enable the Smart Grid. However, current issues for existing radio communications, including this cellular technology, are the requirements of the massive use of ultra-low-latency and reliable connections, bandwidth capacity, as well as security in the Smart Grid use cases just described.

Looking back a few years, almost historically, as reported in [165,166], recent cellular technology known as Narrowband IoT (NB-IoT), which conforms with 5G Industrial IoT, has been approved by 3GPP, published, and tested. It is noted to be the first 5G technology to be used in a public setting. Some Indonesian cellular operators, including Telkomsel and XL, now commercialize NB-IoT as their business service and have started to use it for smart meters [167,168]. By March 2019 the Global Mobile Suppliers Association (GSA) has reported there are 102 operators in 52 countries operating the NB-IoT. This technology is capable of transferring data at 200 kbps and simultaneously connecting up to 100,000 devices in one cell, which meets reliability and density requirements. It is also highlighted that NB-IoT is fit for current 5G and smart grid progress given the higher data rates and frequent communications, not to mention that electric meters are typically in stationary locations in densely populated areas.Lastly, we do not discuss the ultra-low energy features the devices have as they will be closely installed across power lines where the power is easily available.

While 6G is currently in the early stages of research and development and expected to be publicly available by 2030, 5G has been implemented by more than 500 cellular telecommunication operators all over the world (around 8 percent of global mobile connections in 2021) with different degrees of penetration [169,170]. To facilitate machine-to-machine (M2M) green energy data transfer, Low-Power Wide-Area Networks (LPWAN) such as LoRa, ZigBee, and Sigfox have been developed, which offer relatively low bandwidth with various latency, device number, range, and security features. Along with this development, a 5G Industrial IoT called Narrowband IoT (NB-IoT) has been standardized by 3GPP, released, and tested and is considered to be the first 5G technology to go into public service. It coexists in the same networks complementing the existing broadband 5G specially to facilitate low-latency M2M data transfer [171]. Sixth-Generation NB-IoT also has great potential for low-energy narrowband IoT and IIoT communication; the requirements and technologies involved with this are under discussion [172]. There is also a viewpoint that it is unlikely that NB-IoT will be retained as a standalone interface. Instead, 6G would embed similar narrowband, low-power capabilities within more advanced frameworks, such as NOMA-based grant-free access for ultra-massive IoT. These approaches offer greater spectral efficiency, scalability, and energy performance, allowing 6G to address NB-IoT’s core use cases more intelligently and flexibly within its unified mMTC architecture [173].

This 5G LPWAN is capable of transferring data at the level of 200 kbps and simultaneously connecting up to 100,000 devices in one cell, which meets the reliability and density requirements of Smart Grid communication. It is also highlighted that NB-IoT is fit for current 5G and smart grid progress given the higher data rates and frequent communications, not to mention that electric meters are typically in stationary locations in densely populated areas. Finally, we do not discuss the ultra-low energy features the devices have, as they will be closely installed across power lines where the power is easily available.

In broadband cellular technology, the creation of 5G has generally been based on wireless physical- and link-layer enhancements. The key points include massive Multiple-Input Multiple-Output (mMIMO) transceivers, array and smart-beamforming antennas, millimeter-Wave (mmWave) carrier and spectrum aggregation, Non-Orthogonal Multiple Sccess (NOMA), energy efficiency, and backward compatibility [2,174,175,176].

To this end, 3GPP has created a 5G NR (New Radio) standard for universal air interface 5G networks. It defines how devices communicate wirelessly over the air in 5G networks. Enhanced Mobile Broadband (eMBB) is the key use case of 5G NR that focuses on it, delivering significantly improved mobile broadband services. 5G NR is now considered an underlying radio access technology in both 5G standalone (5G SA) and non-standalone (5G NSA) scenarios, aiming to provide higher data rates, improve network capacity and latency, and enhance user experiences compared to previous generations. This includes retaining a dual connectivity protocol that enables devices to connect to both 4G LTE and 5G NR simultaneously, providing a seamless transition for devices moving between coverage areas. There is also a 5G 3GPP standard for Non-Terrestrial Networks (5G NTNs) to accommodate the rest of the world, mainly for rural areas and seas. However, it offers much higher latency due to its long round-trip ground–satellite–ground radio propagation [177,178,179,180]

In the meantime, while there is little exact technical data available for 6G yet, it can be predicted that this would include features that have not been covered in 5G. From [181,182] it can be implied that 6G is expected to address more advanced use cases, such as involving holographic or gadget-free communication and immersive experiences. Compared to 5G, some technical studies estimate that 6G would achieve a data transfer rate of at least 1 Tbps with a latency of less than 100 µs. While 5G connection density has been confirmed to connect 1 million devices per km^2^, 6G is supposed to be 1 billion devices per km^2^. As to the security aspect, 5G is based on given entities of authentication and strong encryption techniques. In contrast, 6G is supposed to apply unknown entities of identification due to the massive use of machine learning and quantum data processing [181,183,184]. By the time the research and standardization process progresses, they will become more defined.

While this research is currently ongoing, each existing wireless generation presents distinct advantages and limitations for IIoT and Smart Grid applications. Pre-5G technologies such as LoRa and ZigBee offer wide coverage and low power use but lack the speed and capacity for critical operations. Fifth-Generation technology provides major improvements to latency, reliability, and device density, supporting real-time control and intelligent energy systems. However, it still faces challenges with energy efficiency and limited AI-native features. Meanwhile, 6G is expected to address these gaps through integrated intelligence, extreme bandwidth, and ultra-dense connectivity. Despite this promise, it must still meet the demands of sustainable operation, consistent reliability, and efficient management of complex, large-scale networks [185,186,187,188,189].

This direction will involve a broad range of advanced technologies such as Intent-Based Networking (IBN), Artificial Intelligence (AI), Distributed Ledger Technology (DLT)/Blockchain (BC), Smart Devices and Gadget-free communication (SDG), Quantum Communication (QC), as well as on-demand network services and applications [181,190]. The 6G use cases are expected to operate smart vital objects such as remote surgery [181], self-driving vehicles, and other, more critical automation [191].

In terms of the Smart Grid, 6G will potentially be used to address at least those categorized in real-time critical and/or high-definition data traffic such as phase measurement, supply–demand resource allocation [191], emergency recovery, and predictive maintenance [192]. Due to the current communication infrastructure constraints, these would be useful for optimizing energy efficiency by minimizing energy waste/loss and maximizing resource utilization of the power grid that would guide us to a greener energy environment. That is how 5G and 6G contribute to making the Smart Grid green. Table 6 compares connection characteristics of the latest wireless technologies commonly or potentially considered in the Smart Grid [193,194,195,196].

### 4.4. Lessons Learned

The 5G applications in smart grids have been briefly investigated in this section. Some critical tasks from the point of view of communication have also been introduced. To give future directions, this section summarizes key performance requirements in the related areas in which technologies could be considered for various Smart Grid applications in the future. We can group the papers investigated into four different focused areas, namely communication in the Smart Grid (CSG), 5G infrastructure in the Smart Grid (5GISG), the Smart Grid (SG) itself, and big data in the Smart Grid (BDSG). Each area has specific key factors that can be pointed out as follows:Sub-millisecond CommunicationAll CSG papers reviewed in this section indicate that, for the most part, the Smart Grid requires millisecond-level communication technology due to the time-critical controls, especially within substat xsion automation systems.Millimeter-Wave BandDiscussions on 5GISG in any technical development are dominated by the concerns of using mmWave to achieve 5G goals (ultra-low latency, more reliability, massive network capacity, increased availability, and a more uniform user experience with more users). Both 5G and 6G use mmWave frequencies that utilize sub-GHz and sub-THz bands. This spectrum will continue to be explored, as these bands offer far higher data rates with ultra-low latency than 4G, which is currently widespread. Semiconductor and antenna designs are the hot topics in this area.Energy Efficiency and ReliabilityIn SG itself, the most fundamental need of a smart grid, which is still under consideration even with the latest advancements, is the reliability and efficiency of energy generation, transmission, and distribution integrated via the electric power grid.Big Data AnalyticsLast but not least is BDSG. As the smart grid produces large volumes of data from IoT devices like smart meters and other sensor networks, it results in big data. Sensors installed in different areas of the smart grid, such as substations and consumer devices, rapidly produce petabytes of data, and it is humanly impossible to analyze this data without Smart Grid big data analytics. Data privacy is also one of the emerging issues intersecting this domain.

The operation of electric power systems is transforming. There is a clear trend towards managing networks at smaller levels with a need for more flexibility and automatic reconfiguration, controlling the interactions of a large number of small groups of users (or even managing them individually), as well as moving towards faster time scales in control and system dynamics. This requires a high-bandwidth, low-latency, and flexible communication system.

Wireless 5G communication meets many of these requirements, and 6G aims to meet them all. Numerous use cases have already been identified and partly tested. Decentralized self-healing protection schemes, grid monitoring based on the large-scale rollout of smart meters, dynamic load production–storage control, peer-to-peer electricity trading, and video surveillance methods. One of the research gaps that still needs attention is sub-ms synchronization for dynamic applications that require PMU-level accuracy. Further research in these areas is essential in order to arrive at reliable and scalable solutions, and cyber-security needs to be a central focus for industry-standard applications.

There are also some implications for reliable Smart Grid operations. In the context of smart grids, the deployment of green communication and control systems must align with strict real-time responsiveness and reliability requirements. Load balancing, fault detection, and distributed energy resource coordination depend on URLLC communication—characteristics that may be compromised when prioritizing energy savings. Here as well, sleep-mode scheduling or low-power protocols can introduce delays or message losses that are unacceptable in grid protection or critical control loops. Additionally, energy-aware routing and edge-based control often require deep integration with operational technologies’ infrastructure, which may be constrained by legacy systems or limited computational resources. Therefore, achieving both energy efficiency and operational dependability in the smart grid domain demands hybrid strategies that consider context-aware trade-offs and include fallback mechanisms to ensure resilience.

## 5. Underexplored Open Challenges and Future Directions

The convergence of beyond-5G and 6G technologies with green IIoT and Smart Grid systems presents several genuinely underexplored research areas. Unlike well-funded domains such as cross-layer optimization, standardization efforts, and AI-driven orchestration, this section identifies research gaps where current knowledge is very limited and future investigation is needed.

### 5.1. TeraHertz Integration Environmental Challenges

The environmental and ecological impacts of deploying THz frequencies in dense 6G networks remain severely underexplored. THz communication, while promising terabit-per-second transmission, faces some environmental challenges that have received limited research attention [199,200]. The high atmospheric absorption of THz signals by water vapor can create unique propagation challenges in outdoor environments, with attenuation levels reaching as high as 104 dB/km under humid conditions [200].

More critically, the environmental implications of widespread THz deployment have not been thoroughly investigated. The need for ultra-dense networks to overcome propagation losses could potentially lead to unprecedented infrastructure density with unknown ecological consequences [201]. Research is clearly needed to assess the cumulative environmental impact of THz emissions on local ecosystems, atmospheric chemistry, and potential disruption to natural communication systems used by wildlife.

The interaction between THz signals and atmospheric constituents through molecular absorption presents both challenges and opportunities for climate sensing applications [199], yet the broader environmental ramifications remain largely unknown.

### 5.2. Behavioral Change and Human Factors in Green Communication Adoption

While technical solutions for energy-efficient communication exist, the human factors influencing adoption are very little researched.

The intersection of technology acceptance models with sustainable communication practices has received minimal attention [202,203]. The role of subjective norms in green communication adoption, as well as the impact of ease-of-operation gaps on sustainable technology use are two important areas to explore better.

The development of behavioral intervention strategies that can effectively promote the adoption of energy-efficient communication practices requires interdisciplinary research combining psychology, technology acceptance theory, and sustainability science [204].

### 5.3. Human–Machine Communication Interfaces in 6G Systems

Future 6G networks are expected to enable unprecedented human–machine interaction through technologies like Extended Reality (XR) and Holographic Telepresence [181,205]. However, fundamental research into how human–machine interface technology systems can be made more sustainable despite their constant connectivity is lacking.

### 5.4. Circular Economy Implementation in 6G Infrastructure

While sustainability is frequently discussed in 6G development, the specific implementation of circular economy principles remains under researched. The integration of lifecycle analysis, material flow optimization, and waste minimization strategies in 6G infrastructure development lacks comprehensive investigation. Research is needed to develop practical frameworks for material recovery, component reuse, and sustainable manufacturing processes in 6G deployment [119].

### 5.5. Quantum-Resistant Security for Energy-Constrained Environments

While quantum communication technologies are advancing, their integration with energy-efficient communication systems faces great challenges. The development of quantum key distribution (QKD) technologies for distributed energy storage systems is a currently emerging area and is supposedly attracting research attention [206].

Furthermore, applied research is needed to develop lightweight quantum-resistant cryptographic protocols that can operate efficiently in energy-constrained environments without compromising security [207,208]. This includes investigating hybrid quantum–classical security schemes, developing energy-efficient quantum key management systems, and creating adaptive security protocols that can scale with varying energy budgets, together with maintaining quantum resistance [209].

The implementation of post-quantum algorithms such as lattice-based cryptography in IoT devices presents unique challenges in terms of computational overhead and energy consumption [210,211]. Research is ongoing to optimize these algorithms for resource-constrained devices while maintaining security against both classical and quantum attacks [212].

### 5.6. Policy Frameworks for Sustainable 6G Deployment

Despite the critical importance of policy in technology adoption, research on policy frameworks specifically designed for sustainable 6G deployment remains limited. The intersection of telecommunications policy with environmental regulation, social equity, and economic development has not been thoroughly investigated [213,214].

It is easy to reveal research gaps regarding adaptive policy mechanisms that can respond to rapidly evolving 6G technologies while ensuring social acceptance and environmental sustainability [215]. This upcoming research effort potentially includes investigating regulatory frameworks for spectrum allocation that consider environmental impact, developing incentive structures for sustainable deployment, and creating cross-sector coordination mechanisms between telecommunications and energy industries.

The development of ethical frameworks for 6G deployment that balance technological innovation with social responsibility represents another underexplored area requiring interdisciplinary research [214]. This, again, means finding and defining governance mechanisms that can ensure equitable access to 6G technologies while minimizing environmental impact.

### 5.7. Context-Aware Fallback Mechanisms for Smart Grid Resilience

While smart grid resilience has received attention, context-aware approaches that adapt to changing environmental, operational, and threat landscapes remain under researched. Current resilience metrics often fail to address the dynamic interplay between geographical location, cyber risk profiles, and varying operational contexts.

Development is clearly needed in terms of context-aware resilience metrics that can quantify a microgrid’s ability to absorb, restore, and adapt to changing circumstances while sustaining critical loads during low-probability, high-impact events. This includes developing adaptive learning mechanisms that adjust based on environmental factors and predictive capabilities that enable better understanding of grid vulnerabilities.

Furthermore, the integration of machine learning algorithms with context-aware systems presents opportunities for developing intelligent fallback mechanisms that can maintain grid stability under varying conditions while optimizing energy efficiency [191].

### 5.8. Lessons Learned

We have identified some genuinely underexplored areas where investigation is needed for the sake of sustainability. Unlike well-funded areas such as cross-layer optimization, standardization efforts, and AI-driven orchestration, these topics require focused research attention to enable the successful deployment of truly sustainable and socially accepted 6G systems.

In this regard, higher priority should be given to interdisciplinary approaches that can address the complex interactions between technology, society, and the environment in the transition to next-generation communication systems. The success of 6G deployment will ultimately depend not only on technical capabilities but also on our ability to address these broader socio-technical challenges.

## 6. Conclusions

The accelerating evolution of wireless communication networks, particularly the transition from 5G to 6G, offers unprecedented opportunities to transform the energy landscape into a more sustainable, intelligent, and resilient ecosystem. This survey has explored the convergence of green communication technologies with the smart grid paradigm, emphasizing the dual objective of reducing the energy footprint of network infrastructures while enabling new capabilities for real-time, data-driven energy management.

The importance of cross-layer optimization and unified architectures has emerged as a key factor for maximizing energy efficiency. The lack of integrated frameworks that cohesively link physical-layer optimizations, network management, edge computing, and AI-based resource management was identified as a major limitation in current research. Additionally, balancing energy savings with the security and reliability demands of critical infrastructures, such as the smart grid, continues to represent a prominent challenge that future solutions must address comprehensively.

Our analysis highlights several enabling technologies that contribute to this vision. Low-power wide-area networks (LPWAN), wake-up radios, and energy harvesting techniques have demonstrated strong potential for supporting battery-less or ultra-low-energy IoT deployments, particularly relevant in distributed Smart Grid scenarios. Edge computing and AI-based orchestration methods provide scalable intelligence close to the data source, helping reduce latency, improve fault detection, and manage distributed energy resources more effectively. On the infrastructure side, optimization of data center operations, network slicing for energy efficiency, and the use of AI to manage traffic load and power allocation in the RAN are emerging as especially important methods.

From the perspective of smart grid applications, this paper has mapped key functions—including state estimation, demand response, load forecasting, and distributed control—to their respective communication requirements. It is evident that while 5G can meet many of today’s latency, reliability, and bandwidth demands, the envisioned features of 6G—such as integrated sensing, ultra-dense edge computing, sub-millisecond latency, and native AI support—will be indispensable for realizing the full promise of future power systems. These include dynamic grid reconfiguration, proactive maintenance via digital twins, and seamless integration of electric vehicles, renewable sources, and prosumers.

However, realizing this vision is not without challenges. The heterogeneity of devices, protocols, and operational requirements across sectors creates substantial interoperability and standardization barriers. Moreover, the environmental gains achieved through improved energy efficiency must not be offset by the increased energy cost of deploying dense and computationally intensive networks. Hence, a lifecycle approach to sustainability—covering design, deployment, operation, and decommissioning—is essential. The development of federated and open test beds, cross-domain collaboration between energy and ICT sectors, and regulation-driven green metrics will play a decisive role in bridging the gap between theory and practice.

While this survey has highlighted technological synergies between beyond-5G, 6G, and green IIoT for smart grids, it is important to recognize that the environmental implications of emerging technologies such as THz communications remain poorly understood. Future research must address how the widespread deployment of THz frequencies impacts atmospheric chemistry and local ecosystems, particularly in ultra-dense network scenarios required for reliable propagation. In addition to optimizing THz communication protocols for energy efficiency and climate sensing, interdisciplinary studies involving environmental science, ecology, and communication engineering are necessary to comprehensively assess and mitigate potential ecological risks.

In addition, the integration of THz and other advanced 6G technologies into smart grid infrastructures requires a sustainability-first design mindset, where the environmental footprint of the lifecycle is evaluated alongside performance metrics. Hence, green networking for Industry 5.0 and smart grids should expand its focus beyond energy savings to include environmental protection as a design constraint. Addressing these unexplored challenges will be essential to ensure that the promise of 6G-enabled smart grids translates into genuine environmental, societal, and industrial benefits without unintended consequences for the the planet.

As we move towards the era of beyond-5G and 6G, a multi-disciplinary research effort is essential to overcome these barriers and utilize the potential of next-generation wireless technologies for sustainable smart grids. Altogether, short- and middle-term future research directions include the development of integrated, cross-layer frameworks, secure and resilient communication protocols, standardized interfaces, and large-scale empirical evaluations. Addressing these challenges and gaps will enable the realization of truly green, intelligent, and reliable energy communication systems, significantly contributing to global sustainability objectives.

## Figures and Tables

**Figure 1 sensors-25-04222-f001:**
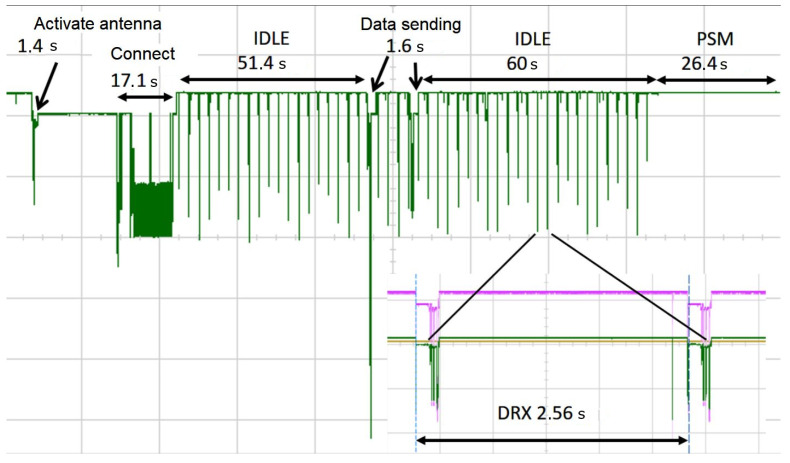
Current draw of a Sara-N2 NB-IoT module during different phases [29].

**Figure 2 sensors-25-04222-f002:**
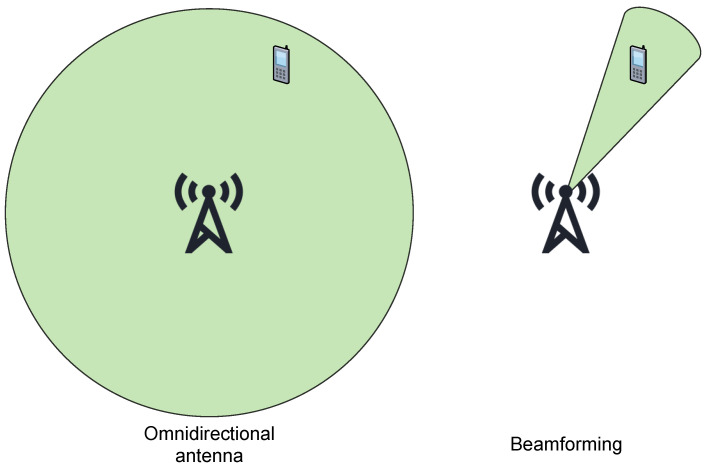
Advantage of beamforming in comparison to traditional omnidirectional antennas.

**Figure 3 sensors-25-04222-f003:**
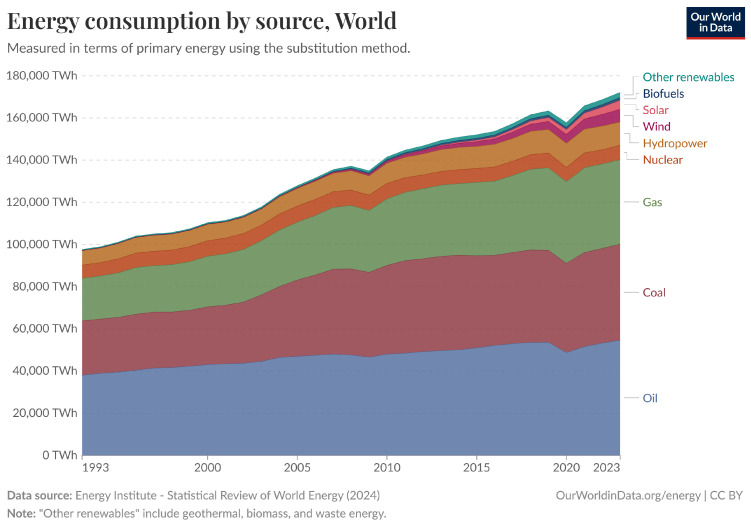
Energy consumption of the world over time—differentiated by energy source [64].

**Figure 4 sensors-25-04222-f004:**
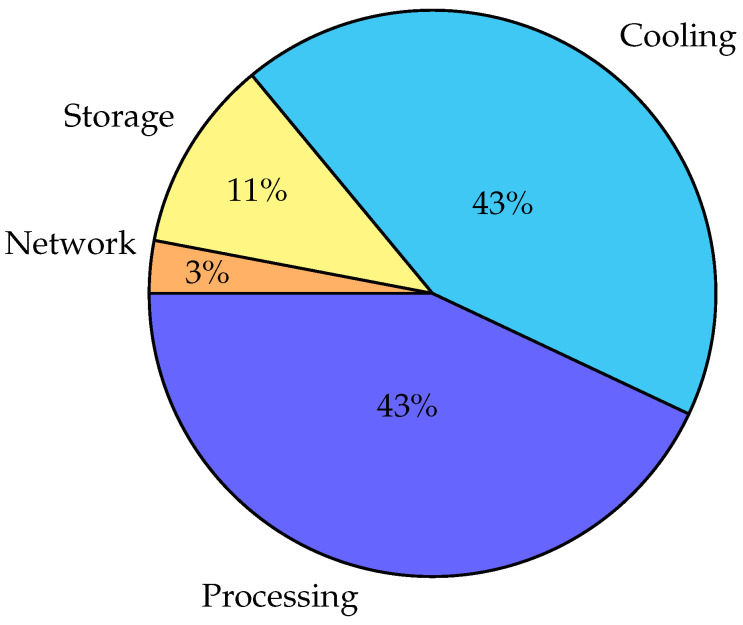
Power consumption distribu tion among data center parts—a typical breakdown [67].

**Figure 5 sensors-25-04222-f005:**
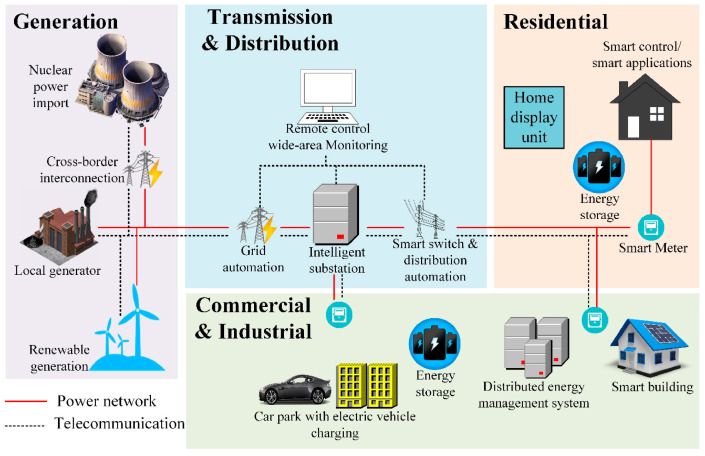
The general architecture of the Smart Grid.

**Table 1 sensors-25-04222-t001:** Comparison of recent surveys: our paper, Maiwada et al., Ezzeddine et al., and Pandiyan et al.

Comparison Criteria	Current Paper	Maiwada et al. (2024) [13]	Ezzeddine et al. (2024) [14]	Pandiyan et al. (2024) [15]
Scope	5G/6G, smart grid, IIoT	5G, Digital twins	AI-based 5G networks	Green IoT and smart grids
Key Focus	Green communications and smart grids	Energy efficiency, DT, and QoS in 5G	Energy efficiency using AI techniques	Sustainable IoT and energy-aware solutions
Reviewed Technologies	LPWAN, MEC, energy harvesting, IoT	Digital twins, DDoS detection, QoS mechanisms	Massive MIMO, NOMA, SDN, NFV, MEC	RFID, Zigbee, BLE, LoRa, MEC, energy-aware protocols
Energy Efficiency Strategies	Integrated cross-layer optimizations	Intrusion detection, QoS-based techniques	ML-driven optimization at multiple network layers	HW-SW co-optimization, harvesting, sleep scheduling
Empirical Validation	Identified gap, recommends empirical studies	Primarily theoretical analysis	Simulation and analytical approaches	High-level review; empirical studies mentioned as future need
Interoperability	Highlighted need, emphasizes cross-domain integration	Not extensively covered	Limited discussion, implied via technology integration	Discussed as critical for 5G-IoT integration
Coverage of Standards	Calls for unified standards	Minimal coverage	Minimal explicit discussion	Explicit discussion of standards
Highlighted Challenges	Fragmentation, security-energy balance, interoperability	DDoS attacks, handover efficiency, QoS consistency	Co-channel interference, network capacity constraints	Heterogeneity, scalability, energy-security trade-off, infrastructure cost

**Table 2 sensors-25-04222-t002:** Comparison of main technical characteristics of SigFox and LoRa.

	SigFox	LoRa
Frequency band	Unlicensed	Unlicensed
Urban range	3–10 km	2–5 km
Rural range	30–50 km	15–20 km
Bandwidth	100 Hz	125 kHz
Maximum data rate	100 bps	50 kbps
Topology	Star	Star
Maximum devices per access point	1 M	100 k
End device cost	<USD 2	USD 3–10
Deployment cost	Subscription-based USD 2–5/year	>USD 100/gateway >USD 1000/base station

**Table 3 sensors-25-04222-t003:** Techniques to reduce energy consumption in 5G networks.

Technique	Description and Benefits
Low-Power Wide-Area Networks (LPWAN)	Enables long-range, low-power communication suitable for IoT applications, significantly reducing the power consumption of end devices.
Wake-Up radios	Special radios are activated only upon receiving specific signals, drastically reducing idle power consumption in IoT devices and sensors.
Mobile Edge Computing (MEC)	Offloading computation-intensive tasks to edge servers close to users, minimizing latency and reducing energy consumption at end devices.
Massive MIMO	Employs large antenna arrays to optimize signal transmission, enhancing spectral efficiency and reducing transmission power per user.
Device-to-Device (D2D) Communication	Enables direct communication between nearby devices, decreasing base station load and energy consumption.
Network Function Virtualization (NFV)	Virtualizing network functions to optimize resource utilization and dynamically allocate resources, thereby reducing overall energy use.
Software-Defined Networking (SDN)	Centralized control allows efficient traffic management and resource allocation, significantly reducing redundant energy consumption.
Heterogeneous Networks (HetNets)	Utilize macro, micro, and pico base stations strategically to manage traffic loads, optimizing energy usage across varying cell sizes.
Dynamic Power Management	Intelligent algorithms dynamically adjust the transmission power of network elements based on real-time demand, saving energy during low-traffic periods.
Energy Harvesting	Integrating renewable sources such as solar panels and RF harvesting to power network equipment, reducing dependence on grid power.
AI-Driven Optimization	Employing AI techniques (ML, DL) for predictive analytics, resource allocation, and dynamic management, enhancing network energy efficiency.
Base Station Sleep Modes	Turning off or putting base stations into low-power states during low usage periods substantially reduces operational energy consumption.
Green Core and Telco Cloud	Optimizing data center components, cooling systems, and employing efficient processors (e.g., ARM-based), significantly lowering core network energy usage.
Cross-layer Optimization	Integrating energy-efficient solutions across multiple network layers (physical, MAC, network), enhancing overall network energy efficiency.
Energy-Efficient Hardware	Deployment of advanced hardware components designed for minimal energy consumption, including power amplifiers, antennas, and processors.

**Table 4 sensors-25-04222-t004:** Summary of the core ideas and approaches appearing in the section.

Approach	Main Papers	Core Ideas
Hardware development	[70,73,78]	Reduction of processor consumption; AgileWatts
Cooling optimization	[82,83,84]	Technological optimization; control policy optimization; high-temperature DC concept
Green energy integration	[87,88,91]	Pre-cooling; thermal storage; geo-distributed load-balancing
Hardware resource optimization	[96,100]	Load-balancing; prediction-based resource allocation
Future possibilities	[115,116]	In-network computing; NOMA and IRS

**Table 5 sensors-25-04222-t005:** Comparison of wireless technologies for IIoT and Smart Grid environments.

Metric	Use Case	Pre-5G	5G	6G (Emerging)
Latency	IIoT	10 ms–1 s	1–10 ms	<1 ms (THz, RIS)
	Smart Grid	10 ms–1 s	<5 ms	Sub-ms (AI control)
Bandwidth	IIoT	0.3–250 kbps	1–100 Mbps, to 10 Gbps	10+ Gbps
	Smart Grid	<50 kbps	10–100 Mbps	10+ Gbps
Device Density	IIoT	1 k–10 k/km^2^	1 M/km^2^	10 M+/km^2^
	Smart Grid	Limited mesh	100 k/km^2^	1 M+/km^2^
Energy Efficiency	IIoT	Ultra-low (10+ yr battery)	Moderate (NB-IoT)	AI-optimized
	Smart Grid	Low-power (LoRa)	Medium (5G PMUs)	RF harvesting
Reliability	IIoT	Low (best-effort)	99.999% (URLLC)	99.99999% (det’istic)
	Smart Grid	Moderate mesh	99.9999%	Self-optimizing
Use Cases	IIoT	Remote monitoring	Predictive maintenance	Autonomous factory
	Smart Grid	AMI, outage alerts	PMUs, pricing	AI grids, DTwins
AI/Edge Support	General	Minimal (centralized)	Basic edge (MEC)	Native AI, RIS

**Table 6 sensors-25-04222-t006:** Typical characteristics of the latest wireless technologies used for IoT access.

Wireless Technology	Use	Data Rate	Latency	Device Number	Range/ Device	Security	Frequency License
LoRa [136]	NAN	∼50 kbps	400 ms	1.5 k	<20 km	built-in	ISM 422, 868, 923 MHz
ZigBee [197]	NAN	∼250 kbps	18 ms	65 k	100 m–1 km	built-in	ISM 2.4 GHz
SigFox [165]	WAN	∼100 bps	>400 ms	50 k	10–40 km	none	ISM 433, 868, 915 MHz
5G NB-IoT [165,166]	NAN, WAN	200 kbps	0.6–1 s	100 k	1–10 km	built-in	Cellular 5G Standard
5G NB2-IoT [198]	NAN, WAN	∼700 kbps	0.6–1 s	1000 k	10 km	built-in	Cellular 5G Standard
5G NTN [177,178,180]	WAN	varied	long	multitude	wide	built-in	5G NTN Standard
5G NR [181,182]	HAN, NAN, WAN	∼20 Gbps	<1 ms	1 M/km^2^	<1 km	known entities	Cellular 5G Standard
6G [183,184]	HAN, NAN, WAN	∼1 Tbps	10–100 µs	10 M/km^2^	<1 km	unknown entities	Cellular 6G Standard

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
