# Peer review of "How Beyond-5G and 6G Makes IIoT and the Smart Grid Green—A Survey"

_sensors, 2025, doi:10.3390/s25134222_

Round 1
Reviewer 1 Report
Comments and Suggestions for Authors
This article provides a recent and comprehensive review of how the development of beyond-5G (B5G) and 6G communications technologies promotes energy efficiency and sustainability in smart grid and IIoT systems. The article is rich in technical content, well structured, and well cited. Nevertheless, to contribute academic weight and be worthy of publication in the Sensors journal, some critical areas of enhancement need to be addressed in terms of clarity, organization, methodological intricacy, and scientific value.
- Though the paper is fairly general in scope over technologies and approaches, it does not propose a unique taxonomy, framework, or grand vision. Propose a new architecture or classification that comprehensively maps enabling technologies (e.g., LPWAN, MEC, wake-up radios) to smart grid applications (e.g., FLISR, MLC, DG) over multiple communication and latency paradigms. A graphical conceptual model would greatly benefit reader comprehension.
- Many sections cover existing technologies (e.g., LoRa vs. Sigfox, D2D, MEC) but stop short of going the complete distance of making serious analytical comparisons (e.g., trade-offs, gaps in technology, performance baselines). Give performance comparison tables (latency, energy savings, bandwidth, device scalability, etc.) with quantitative values when available. Clearly distinguish the use of each technology for IIoT versus smart grid environments.
- The paper does not adequately address the challenges, limitations, or undesirable side effects of employing green technologies (e.g., security vs. energy trade-off, MEC control overhead, edge deployment expenses). Add a subsection or table highlighting existing limitations and technical challenges of each of the considered approaches, along with mitigation strategies.
- Although the title is B5G/6G, the bulk of the analysis still remains focused on 5G or overall green IoT approaches. Expand discussion of 6G-enabling technologies like RIS, NOMA, and quantum communication. How does 6G distinguish itself in terms of energy-efficient design and smart grid integration? What are the likely deployment timelines and regulatory issues?
- The narrative tends to recycle background information (e.g., benefits of MEC or PMUs), cross sections, reducing conciseness and scientific significance. Edit for conciseness. Consolidate redundant information and emphasize unique points in each subsection.
- Some sections contain awkward phrasing (e.g., "there are networks where one node generates small, periodic information").
- Suggest thorough proofreading to exclude typos (e.g., "arount 15.4 billion"), subject-verb disputes, and improve scholarly tone.
- Some of the numbers (e.g., Fig. 1, Fig. 4) would be clearer with improved captions and being placed within the text. Make sure that each is clearly labeled and discussed.
- Tables 3 and 4 are useful, but they could be made even more useful by including references for the data and compact analytical findings.
- I recommend citing additional references such as [https://doi.org/10.3390/en13112762] to strengthen and expand the discussion.
Author Response
Thanks a lot for the reviewer insights. We marked the issues raised as ***, and our answers +++.
Please note that we really improved the paper significantly. Length changed from 30 to 37 pages, citations from 168 to 205, and we added several tables for comparison. We hope that you will see the expected improvement in the manuscript.
*** Though the paper is fairly general in scope over technologies and approaches, it does not propose a unique taxonomy, framework, or grand vision.
Propose a new architecture or classification that comprehensively maps enabling technologies (e.g., LPWAN, MEC, wake-up radios) to smart grid applications (e.g., FLISR, MLC, DG) over multiple communication and latency paradigms. A graphical conceptual model would greatly benefit reader comprehension.
+++ We have not aimed to propose taxonomies or architectures, this was not the scope, as the paper is of a survey kind. We showed the complexity of the problem, and added further tables to underline this.
*** Many sections cover existing technologies (e.g., LoRa vs. Sigfox, D2D, MEC) but stop short of going the complete distance of making serious analytical comparisons (e.g., trade-offs, gaps in technology, performance baselines). Give performance comparison tables (latency, energy savings, bandwidth, device scalability, etc.) with quantitative values when available.
+++ We improved this part, although not comprehensively. We cited papers with exact measures. Thanks for the suggestion.
*** Clearly distinguish the use of each technology for IIoT versus smart grid environments.
+++ These are now distinguished in a chapter level
*** The paper does not adequately address the challenges, limitations, or undesirable side effects of employing green technologies (e.g., security vs. energy trade-off, MEC control overhead, edge deployment expenses).
+++ We have added subsections, thanks for this suggestion!
*** Add a subsection or table highlighting existing limitations and technical challenges of each of the considered approaches, along with mitigation strategies.
+++ We have added subsections, thanks for this suggestion!
*** Although the title is B5G/6G, the bulk of the analysis still remains focused on 5G or overall green IoT approaches. Expand discussion of 6G-enabling technologies like RIS, NOMA, and quantum communication.
+++ Thanks for this important note, we have added a complete subsection with proper references to tackle these.
*** How does 6G distinguish itself in terms of energy-efficient design and smart grid integration? What are the likely deployment timelines and regulatory issues?
The narrative tends to recycle background information (e.g., benefits of MEC or PMUs), cross sections, reducing conciseness and scientific significance.
+++ Thanks, we addressed now all three issues, 6G specialities (energy efficiency is a must), 6G timelines, and the smart grid integration.
*** Edit for conciseness. Consolidate redundant information and emphasize unique points in each subsection.
+++ We edited the paper and aimed for this, however, some redundancy is inevitable.
*** Some sections contain awkward phrasing (e.g., "there are networks where one node generates small, periodic information").
+++ We corrected the language and phrases throughout.
*** Suggest thorough proofreading to exclude typos (e.g., "arount 15.4 billion"), subject-verb disputes, and improve scholarly tone.
+++ We corrected the language and phrases throughout.
*** Some of the numbers (e.g., Fig. 1, Fig. 4) would be clearer with improved captions and being placed within the text. Make sure that each is clearly labeled and discussed.
+++ Thanks. We improved the captions in many places.
*** Tables 3 and 4 are useful, but they could be made even more useful by including references for the data and compact analytical findings.
I recommend citing additional references such as [https://doi.org/10.3390/en13112762] to strengthen and expand the discussion.

Reviewer 2 Report
Comments and Suggestions for Authors
The observations are as follows
- The authors should clearly mention the motivation behind this study.
- It would be better if the authors highlighted the key contributions.
- The uniqueness of this study can be established through a proper and in-depth analytical comparison between the existing review/survey papers and this paper. The authors should include such a comparative table.
- Although the authors have presented different techniques to reduce energy consumption within 5G cellular networks, additionally a comprehensive table will be a better choice.
- The authors should have a section addressing briefly about the 5G and 6G network architectures, and IIoT ecosystems.
- Also, the author should highlight the Green Communication and Sustainability metrics and KPIs in Green IoT.
- As the authors have targeted the 6G for the IIoT, the coverage of the discussion can be further extended to Edge and Fog Computing, AI/ML for Energy Optimization, Reconfigurable Intelligent Surfaces (RIS), Digital Twins & Sustainable Automation.
- Add open challenges and future directions.
- After the conclusion, all the statements need to be properly framed.
Author Response
Thanks a lot for the reviewer insights. We marked the issues raised as ***, and our answers +++.
Please note that we really improved the paper significantly. Length changed from 30 to 37 pages, citations from 168 to 205, and we added several tables for comparison. We hope that you will see the expected improvement in the manuscript.
*** The authors should clearly mention the motivation behind this study.
+++ We have added a clarified motivational paragraph to the introduction.
*** It would be better if the authors highlighted the key contributions.
+++ We have added an itemized list highlighting the key contributions.
*** The uniqueness of this study can be established through a proper and in-depth analytical comparison between the existing review/survey papers and this paper. The authors should include such a comparative table.
+++ Thanks for this request. We have added such a comparison table.
*** Although the authors have presented different techniques to reduce energy consumption within 5G cellular networks, additionally a comprehensive table will be a better choice.
+++ Thanks for this request. We have added such comparison table.
*** The authors should have a section addressing briefly about the 5G and 6G network architectures, and IIoT ecosystems.
+++ We have cited works that do that. The paper would be extraordinarily long, but by adding textbook knowledge it may be boring for experts.
*** Also, the author should highlight the Green Communication and Sustainability metrics and KPIs in Green IoT.
+++ Thanks for this note, we added a related paragraph on KPIs and metrics with appropriate citation.
*** As the authors have targeted the 6G for the IIoT, the coverage of the discussion can be further extended to Edge and Fog Computing, AI/ML for Energy Optimization, Reconfigurable Intelligent Surfaces (RIS), Digital Twins & Sustainable Automation.
+++ We partially addressed all of these, although not putting them in focus because that could be another huge paper.
*** Add open challenges and future directions.
We have added these to the conclusions.
*** After the conclusion, all the statements need to be properly framed.
+++ This is also done, Thanks.

Reviewer 3 Report
Comments and Suggestions for Authors
- The tables (e.g., Table 3 and Table 4) are dense, cluttered, and not reader-friendly in the manuscript. The formatting is inconsistent, and the tabular comparisons lack visual clarity, making it difficult to extract valuable information. Use simplified, well-structured comparative tables with fewer entries per table. Using bullet points or symbols, highlight key methodological differences, outcomes, and limitations. Add captions that explain what the reader should learn from each table.
The authors should explain a 5G and 6G communication technologies framework, including types of authentication, energy efficiency, technology used, application, and evaluated risk. They should also try to use a conceptual diagram or workflow to differentiate between existing solutions and identify future research pathways.
-While Green IoT in 5G and 6G networks is discussed in detail, other relevant technologies like edge AI or lightweight protocols for IIoT are not present in the manuscript. The authors should broaden the scope slightly to address lightweight mechanisms, energy-aware protocols, or trust architectures, which are emerging in IIoT
Author Response
Thanks a lot for the reviewer insights. We marked the issues raised as ***, and our answers +++.
Please note that we really improved the paper significantly. Length changed from 30 to 37 pages, citations from 168 to 205, and we added several tables for comparison. We hope that you will see the expected improvement in the manuscript.
***- The tables (e.g., Table 3 and Table 4) are dense, cluttered, and not reader-friendly in the manuscript. The formatting is inconsistent, and the tabular comparisons lack visual clarity, making it difficult to extract valuable information.
++++ We have amended the tables, with limited success. This is an editorial issue; if the Template or the publisher allows changes to the default settings they should suggest.
*** Use simplified, well-structured comparative tables with fewer entries per table. Using bullet points or symbols, highlight key methodological differences, outcomes, and limitations.
+++ We do not understand the purpose of this request, sorry... We aimed to create the comparison tables exactly for allowing complex analysis. Fewer entries per table goes against this will. The differences, outcomes and limitations are hard to include altogether in the tables, so we "outsourced" these to the lessons learned section of each chapter.
***Add captions that explain what the reader should learn from each table.
+++ Thanks for this note, we improved the captions - hopefully the improvement appear significant. We also improved the captions for certain Figures.
*** The authors should explain a 5G and 6G communication technologies framework, including types of authentication, energy efficiency, technology used, application, and evaluated risk.
+++ Thanks for this suggestion. We applied this related to energy efficiency and technologies used - but authentication, application, and evaluated risk is out of the focus of this paper, as it focuses on the junction of 5G/6G, IIoT and smart grid/ green solutions. The mentioned concepts - from authentication to application is really a far stretch, with all due respect. We now added references to 5G and 6G architectures for readers already in the Introduction. In our opinion, their architecture itself has very little effect of IIOT. The effects on energy are described in the core and telco cloud parts, because they are majorly relevant there.
***They should also try to use a conceptual diagram or workflow to differentiate between existing solutions and identify future research pathways.
+++ We added an extended future research section to the conclusion.
*** While Green IoT in 5G and 6G networks is discussed in detail, other relevant technologies like edge AI or lightweight protocols for IIoT are not present in the manuscript. The authors should broaden the scope slightly to address lightweight mechanisms, energy-aware protocols, or trust architectures, which are emerging in IIoT
+++ Thanks, we briefly addressed these, as well as highlighting a survey paper that directly deals with these.

Reviewer 4 Report
Comments and Suggestions for Authors
The keyword “sustainability”, is not addressed well in the body of the paper.
The IIOT is not discussed in the body of the paper. The elaboration is only on the IoT !!
In Line 405, the reference should be corrected.
Will we have 6G NB-IoT for IoT devices? This is not mentioned in the body of the paper. In my opinion, the authors should take into account that each IoT device requires Narrow Bandwidth and generate low data rate, then when taking about 5G and 6G then this means 5G NB-IoT and 6G NB-IoT.
Comments on the Quality of English LanguageShould be improved
Author Response
Thanks a lot for the reviewer insights. We marked the issues raised as ***, and our answers +++.
Please note that we really improved the paper significantly. Length changed from 30 to 37 pages, citations from 168 to 205, and we added several tables for comparison. We hope that you will see the expected improvement in the manuscript.
***The keyword “sustainability”, is not addressed well in the body of the paper.
+++ Thanks for pointing it out! We included coverage of sustainability towards 6G, and also for industry4.0 and 5.0.
***The IIOT is not discussed in the body of the paper. The elaboration is only on the IoT !!
+++ We provide an introduction of IIoT erms now in Chapter 2, and then reflect on IIoT in various places in the improved manuscript.
***In Line 405, the reference should be corrected.
+++ Thanks - we have corrected this.
*** Will we have 6G NB-IoT for IoT devices? This is not mentioned in the body of the paper. In my opinion, the authors should take into account that each IoT device requires Narrow Bandwidth and generate low data rate, then when taking about 5G and 6G then this means 5G NB-IoT and 6G NB-IoT.
+++ Thanks for this note, we included 6G NB-IoT prospects with the appropriate citation beside 5G NB-IoT.

Round 2
Reviewer 2 Report
Comments and Suggestions for Authors
The authors have adequately addressed almost all the comments I raised in my previous review and revised the manuscript. However, I still believe that a separate section needs to be added to accommodate the open challenges and future directions.
Author Response
Thanks for the note.
We have added a completely new chapter (#5) addressing the open challenges in the long run, taking into account socio-economic and sustainability issues as well. Regarding open challenges per the original chapters, we included those in the related individual lessons learned sections.
We hope these fulfill the expectations.
Reviewer 3 Report
Comments and Suggestions for Authors
After reviewing the revisions made by the authors, I see that the article is of high quality. I recommend accepting the article.
Author Response
Thanks for the kind review!